# A Single DNA Binding Site of DprA Dimer Is Required to Facilitate RecA Filament Nucleation

**DOI:** 10.3390/ijms26167873

**Published:** 2025-08-15

**Authors:** Irina Bakhlanova, Begoña Carrasco, Aleksandr Alekseev, Maria Yakunina, Natalia Morozova, Mikhail Khodorkovskii, Michael Petukhov, Dmitry Baitin

**Affiliations:** 1Petersburg Nuclear Physics Institute Named by B.P. Konstantinov of National Research Center «Kurchatov Institute», Orlova Roscha 1, 188300 Gatchina, Russia; 2Department of Microbial Biotechnology, Centro Nacional de Biotecnología, CNB-CSIC, 3 Darwin Str., 28049 Madrid, Spain; 3Research Center of Nanobiotechnologies, Peter the Great St. Petersburg Polytechnic University, Polytechnicheskaya, 29, 195251 St. Petersburg, Russia; 4National Research Center «Kurchatov Institute», Akademika Kurchatova pl. 1, 123182 Moscow, Russia

**Keywords:** RecA, Dpra, protein–protein interactions, ATP/dATP hydrolysis, nucleoproteins

## Abstract

DprA (also known as Smf) is a conserved RecA mediator originally characterized by its role in natural chromosomal transformation, yet its widespread presence across bacteria hints at broader DNA metabolic functions. Here, we demonstrate that *Bacillus subtilis* DprA enhances the frequency of *Escherichia coli* Hfr conjugation in vivo. In vitro, RecA·ATP binds and cooperatively polymerizes in a 50-nucleotide (nt) polydeoxy T (dT)_50_ ssDNA to form dynamic filaments that SSB inhibits, an effect fully reversed by *Bacillus subtilis* DprA. *Escherichia coli* RecA bound to (dT)_21_ exhibits minimal dATPase activity, but the addition of *B. subtilis* DprA significantly stimulates RecA dATP hydrolysis. *B. subtilis* RecA·dATP readily assembles on (dT)_20_ complexes, and DprA allosterically activates RecA on even shorter (dT)_15_ substrates. Combining biochemical assays with a fully atomic model of the RecA–DprA–ssDNA complex, we proposed that only one DNA binding site of the DprA dimer engages the ssDNA during RecA loading, owing to steric constraints. This work refines the mechanism of DprA-mediated RecA nucleation and defines the minimal ssDNA footprint required for mediator activity.

## 1. Introduction

DprA is a ubiquitous, evolutionarily conserved protein that directly interacts with RecA and single-strand binding proteins (SsbA and SsbB) to facilitate natural chromosomal transformation, as supported by yeast two-hybrid, pull-down, and Förster resonance energy transfer data [1,2]. Upon entry of exogenous single-stranded DNA (ssDNA) into the cytosol, SsbA/B coat the incoming single-stranded DNA, blocking nonproductive binding. DprA dimers then partially displace the SsbA/B and load RecA onto ssDNA, acting as mediator proteins to initiate natural chromosomal transformation [3]. DprA’s role as a natural chromosomal transformation factor in numerous Firmicutes species has been well-established, with extensive studies on *Bacillus subtilis* (DrpA*_Bsu_*) and *Streptococcus pneumoniae* (DrpA*_Spn_*) [2,4,5]. In *Streptococcus pneumoniae,* RecA (RecA*_Spn_*) further destabilizes the DprA*_Spn_* dimer interface to complete ssDNA hand-off, underscoring a finely tuned mediator mechanism [4]. Across Firmicutes and other phyla, high sequence conservation and the ability to bind both single- and double-stranded DNA support a critical, yet still underexplored, cellular function of the protein [2,5,6]. Biochemical studies have revealed at least three discrete activities of DprA proteins: (a) displacement of SsbA/B “roadblocks” and positive mediation of RecA filament assembly on ssDNA [5], (b) promotion of strand annealing between complementary Ssb-coated ssDNAs [7], and (c) counteracting function of negative RecA modulator RecX [8].

Moreover, in *S. pneumoniae*, DprA*_Spn_* coordinates the shut-off of competence to prevent deleterious effects of prolonged DNA uptake [9]. In vitro reconstitution confirmed that pneumococcal DprA*_Spn_* displaces SSB from ssDNA and stimulates RecA*_Eco_* nucleation and growth on mixed DprA*_Spn_*-ssDNA-SSB complexes [2], demonstrating heterospecific interaction across distant bacterial proteins.

Although first characterized in naturally competent species, DprA homologs are widespread—even in bacteria lacking known competence pathways—hinting at additional roles in DNA metabolism beyond transformation [10,11]. For example, overexpression of *Escherichia coli* DprA (DprA*_Ec_*/Smf*_Ec_*) restores chromosomal transformation in competent *Haemophilus influenza* mutants [12], indicating biologically significant cross-species RecA–DprA compatibility.

Structural analyses of DprA from Gram-positive and Gram-negative bacteria illuminate its multifunctionality. DprA*_Spn_* comprises an N-terminal sterile alpha motif (SAM) and a C-terminal Rossmann fold that assemble into tail-to-tail dimers, with overlapping surfaces for DprA–DprA and DprA–RecA interactions. In contrast, *Helicobacter pylori* DprA (DprA*_Hpy_*) features an N-terminal Rossmann fold and C-terminal winged-helix domain (DML1) yet retains conserved interfaces for ssDNA and RecA binding [13,14].

The DprA*_Eco_* protein comprises 374-amino acid residues. While the crystal structure of full-length DprA*_Eco_*, which contains SAM, RF, and DML1 domains, remains unresolved, the structure of its DprA*_Spn_* homolog has been elucidated [13]. DprA*_Spn_* forms tail-to-tail dimers via the association of a sterile alpha motif domain and a Rossmann fold. Structural mapping of key interacting residues within DprA*_Spn_* indicates an overlap between the DprA*_Spn_*–DprA*_Spn_* and DprA*_Spn_*–RecA*_Spn_* interaction surfaces [13]. Overall, these features suggest a competitive interaction between DprA*_Spn_* and a truncated variant of RecA*_Spn_* lacking the first 27 residues [4]. Multiple pieces of in vitro evidence have demonstrated heterospecific interaction between DprA*_Spn_* and non-cognate RecA proteins, raising the possibility of using DprA as a universal protein across species boundaries [2]. To assess whether a Firmicutes DprA is functional during Hfr conjugation, we examined the activity of plasmid-expressed DprA from *B. subtilis* (DprA*_Bsu_*). DprA*_Bsu_* was selected due to its well-characterized interaction with RecA*_Bsu_*, as confirmed by yeast two-hybrid assays and fluorescence resonance energy transfer (FRED) experiments [1]. In contrast, a binary interaction between full-length DprA*_Spn_* and RecA*_Spn_* was undetectable via yeast two-hybrid assays [13]. Interestingly, a truncated RecA*_Spn_* variant lacking its N-terminal domain responsible for monomer-monomer interactions does bind full-length DprA*_Spn_* [13].

However, the function of DprA in non-transformable bacteria remains unclear. Here, we investigated the ability of DprA*_Eco_* to substitute for endogenous Smf/DprA*_Bsu_* during *Escherichia coli* Hfr conjugation and measured its impact on homologous recombination in vivo. In parallel, we employed defined oligodeoxythymidine substrates and ATP/dATP hydrolysis assays to dissect the molecular mechanism of DprA-mediated RecA loading. Finally, we built an atomic-level model of the RecA–DprA–ssDNA assembly to explain how steric constraints restrict active DNA engagement to a single protomer within the DprA dimer.

## 2. Results and Discussion

### 2.1. DprA Significantly Enhances Recombinational Exchange Frequency (FRE)

Conjugation continuously introduces new ssDNA that is transferred into recipients during Hfr mating, with markers transferred sequentially (leu^+^→ara^+^→thr^+^) according to the Haldane model of recombination [15]. FRE (frequency of recombinational exchanges) was defined as the average number of crossover events per *E. coli* genome equivalent per 100 min. Over the past 20 years, an assay capturing essentially all genetic exchanges during Hfr mating was developed to measure FRE. This approach aligns with earlier methods based on recombinant yield and marker–marker linkage [16].

At least two quantitative parameters have been utilized to characterize conjugational recombination—the yield of recombinants and the linkage of two transferred donor markers. For wild-type *E. coli*, FRE = 5.0. [17,18]. ΔFRE is calculated as FRE_1_/FRE_2_, where FRE_1_ is the FRE of the test and FRE_2_ that of the reference. To assess DprA’s role, we conducted Hfr conjugation experiments with IPTG-induced expression of either native DprA*_Eco_* or heterologous DprA*_Bsu_*. Overexpression timing was optimized to avoid metabolic overload and approximate physiological levels. At 37 °C, DprA*_Eco_* and DprA*_Bsu_* increased FRE by 2.6- and 6.4-fold relative to vector alone (Table 1a), indicating that DprA*_Bsu_* is a stronger recombination activator under these conditions. Lowering the induction temperature to 30 °C further amplified FRE 17- to 19-fold for DprA*_Eco_* and DprA*_Bsu_*, respectively (Table 1b).

As previously described, FRE values in the wild-type background increased in a temperature-dependent manner, showing a 2-fold rise at 42 °C [16]. This unexpected trend underscores the robustness of the observed effect. This suggests that temperatures as high as 37 °C inhibit intracellular protein abundance or activity of proteins or both. Collectively, these data demonstrate that DprA acts as a potent positive regulator of HR in Gram-negative bacterial conjugation and that DprA*_Bsu_* efficiently substitutes for the endogenous mediator in *E. coli* cells.

### 2.2. DprA_Bsu_ Promotes SSB_Eco_ Displacement from ssDNA

To date, there are no published reports of biochemical purification of DprA*_Eco_*, likely due to its susceptibility to proteolytic degradation and poor solubility in aqueous media. The observation of heterospecific interactions between DprA*_Bsu_* and RecA*_Eco_*, combined with the high degree of conservation between RecA*_Bsu_* and RecA*_Eco_*, led us to use RecA*_Eco_* to investigate the effect of DprA*_Bsu_* binding.

RecA filament assembly on SSB–ssDNA complexes can be monitored indirectly via ATP hydrolysis, and DprA*_Bsu_* has been validated as a mediator of this process.

To test whether DprA*_Bsu_* facilitates RecA*_Eco_* nucleation on SSB-coated ssDNA, we preincubated circular M13mp8 ssDNA (12 μM) with SSB*_Eco_* protein (0.5 μM) to achieve full coverage. Sub-stoichiometric RecA*_Eco_* (1 μM) and increasing concentrations of DprA*_Bsu_* were then added, and ATP hydrolysis was recorded. In the initial minutes, in the absence of DprA*_Bsu_*, RecA*_Eco_* exhibited a prolonged lag phase of ATP hydrolysis, indicating a temporary delay in filament nucleation due to the obstacle presented by SSB. Addition of DprA*_Bsu_* not only eliminated this lag but also increased the steady-state ATPase rate in proportion to DprA*_Bsu_*, indicating active SSB*_Eco_* displacement and efficient RecA*_Eco_* loading (Figure 1a). The maximal RecA*_Eco_* ATPase rate was reached at 1 DprA*_Bsu_* monomer per 150-nt of DNA.

On circular ssDNA, RecA*_Eco_* gradually displaces SSB*_Eco_*, whereas on linear ssDNA with a free 5′ end, SSB*_Eco_* more effectively dislodges RecA*_Eco_*, reflecting the 5′→3′ polarity of RecA*_Eco_* filament disassembly [19,20].

The dynamic equilibrium between SSB*_Eco_* and RecA*_Eco_* on ssDNA depends on their concentrations, the order of addition, and DNA topology. To focus strictly on direct protein–protein interactions, we used a linear, unstructured (dT)_50_ oligonucleotide. To directly assess DprA*_Bsu_*’s ability to overcome SSB*_Eco_*-mediated inhibition, circular ssDNA was replaced by a linear 50-nt poly(dT). RecA*_Eco_* (3 µM) was present in excess over dT_50_ (5 µM in nt), ensuring full substrate binding. When RecA*_Eco_* was preincubated with (dT)_50_ and ATP, filaments formed rapidly and hydrolyzed ATP at a steady state rate near values ~25–30 min^−1^ (Figure 1b).

Addition of SSB*_Eco_* (0.5 µM) to preformed RecA*_Eco_*-DNA complexes rapidly displaced RecA*_Eco_*, abolishing ATPase activity (Figure 1b). Strikingly, the addition of DprA*_Bsu_* (80 nM) completely restored ATP hydrolysis to its original rate, even when SSB*_Eco_* was pre-bound. The kinetics returned to the same steady state, demonstrating complete and reversible SSB*_Eco_* displacement. Because dT_50_ lacks secondary structure, these effects are attributable to specific protein–protein interactions. Two non–mutually exclusive mechanisms may account for DprA*_Bsu_*-mediated RecA*_Eco_* loading: (i) DprA first binds ssDNA, displaces SSB*_Eco_* via direct interaction, and thereby creates a platform for RecA nucleation [5,21]; (ii) DprA*_Bsu_* and RecA*_Eco_* cooperatively interact to enhance RecA*_Eco_*’s nucleation frequency on SSB-coated ssDNA, irrespective of which protein binds DNA first.

### 2.3. DprA_Bsu_ Facilitates RecA Nucleation onto Short (dT_n_) Oligos

To know how DprA*_Bsu_* influences RecA*_Eco_* filament formation on oligonucleotides of defined length, experiments were conducted in which RecA*_Eco_* was added prior to or following DprA*_Bsu_*. RecA*_Eco_*’s affinity for ssDNA increases with substrate length, yielding stable filaments on long oligos but failing on shorter ones [22,23]. Varying the substrate length helps isolate the nucleation phase from filament extension. Given the cooperative nature of monomer binding, an insufficient number of subunits in the filament compromises the complex stability. The other current view is that the nucleation occurs as a 5-subunit oligomer or even a dimer, followed by filament growth on ssDNA [24,25]. Short oligonucleotides, such as (dT)_30_ to (dT)_40_, proved to be poor substrates for RecA-mediated ATP hydrolysis and dynamic filament formation. Protein interaction with such limited-length DNA results in only modest ATP hydrolysis. However, increasing ssDNA length gradually enhances RecA-mediated hydrolysis, reaching peak levels. A (dT)_34_ oligo can accommodate 10–11 RecA monomers. Due to the dynamic between association and dissociation, the actual number of bound monomers may be lower at any given moment. At the concentration of RecA protein used in our experiment, ATP hydrolysis is activated by less than 20% of the potential maximum value (Figure 2a). We pre-incubated 3 µM RecA*_Eco_* with (dT)n (5 µM in nt) in ATP and measured ATP hydrolysis. Short (dT)_30_–(dT)_40_ supported only modest activity, whereas (dT)_34_, capable of binding ~10–11 RecA*_Eco_* monomers, activated less than 20% of the maximal ATPase rate (Figure 2a). At this concentration, RecA*_Eco_* nucleation and filament growth are severely limited by substrate length. The addition of 0.06 μM DprA*_Bsu_* dramatically elevated RecA*_Eco_* ATPase rates on (dT)_34_, regardless of protein addition order. Enhancement was linear with DprA*_Bsu_* concentration and reached saturation at a 1 DprA*_Bsu_* dimer/(dT)_34_ ratio (0.24 μM DprA*_Bsu_*), implying a single mediator dimer engages each DNA oligo [13].

To define the minimal substrate length required for DprA*_Bsu_*-mediated stimulation of RecA*_Eco_* nucleation, ATP was replaced by dATP, which increases RecA*_Eco_*·ssDNA stability and nucleation frequency [26,27].

Switching from ATP to dATP allowed us to define the minimum site size required for DprA*_Bsu_*-mediated RecA loading. Under dATP, RecA*_Eco_* binds ssDNA with higher affinity and slower disassembly, enhancing nucleation events. RecA*_Eco_*·dATP on (dT)_34_ hydrolyzed dATP efficiently (Figure 2b), confirming sufficient filament assembly. We then assayed dATP hydrolysis on oligos ranging from (dT)_50_ to (dT)_21_ in the presence or absence of DprA*_Bsu_* (0.3 µM DprA) (Figure 2b). On (dT)_50_, RecA*_Eco_*·dATP activity was near-maximal, obscuring any mediator effect. As oligo length decreased, DprA*_Bsu_*’s stimulatory role became increasingly evident, with a 7- to 8-fold activation on (dT)_21_. RecA*_Eco_*·dATP alone showed negligible hydrolysis on (dT)_21_, but DprA*_Bsu_* fully restored robust activity, indicating that, within the examined range, a 21-nt ssDNA (≈seven RecA protomers) is the minimal site for mediator-assisted nucleation. (Figure 2b).

### 2.4. DprA_Bsu_ Promotes Dynamic RecA_Bsu_·dATP Filaments on Short ssDNA

RecA*_Bsu_* exhibits no detectable dATPase activity in the absence of ssDNA, and DprA*_Bsu_* lacks intrinsic dATPase function [7]. Thus, all observed dATP hydrolysis reflects RecA*_Bsu_*·dATP nucleation on the provided oligonucleotides. To determine the minimum substrate length required for DprA to stimulate RecA nucleation, experimental conditions were optimized using homologous proteins and dATP. The nucleotide length required for DprA*_Bsu_* to stimulate RecA*_Bsu_*·dATP nucleoprotein filament formation was measured.

RecA*_Bsu_*·dATP (1 RecA monomer/2-nt (dT)_n_) binds to and cooperatively polymerises on (dT)_30_ ssDNA, forming a dynamic filament that hydrolyzes dATP with a k_cat_ of 2.3 ± 0.3 min^−1^ (Figure 3). In contrast, with (dT)_20_ and (dT)_15_ ssDNA substrates, RecA*_Bsu_*·dATP showed significantly reduced hydrolysis rates (k_cat_ = 0.2 ± 0.02 min^−1^ and 0.19 ± 0.02 min^−1^, respectively), only slightly above background levels observed with RecA or DprA in the absence of ssDNA (-ssDNA).

To test whether RecA*_Bsu_*·dATP undergoes a functional transition upon interaction with DprA*_Bsu_*, and whether short DNA substrates might confer structural stability for RecA loading, DprA*_Bsu_* was added to the reaction mixture. Addition of DprA_Bsu_ not only restored but enhanced dATP hydrolysis across all lengths, yielding k_cat_ values of 4.4 ± 0.4 min^−1^, 2.0 ± 0.4 min^−1^ on (dT)_20_, and (dT)_15_. The effect of DprA*_Bsu_* on maximizing RecA*_Bsu_*-mediated dATP hydrolysis with (dT)_20_ or 1.1 ± 0.2 min^−1^ on (dT)_15_ corresponded to an 8-fold and 10-fold activation on the shortest substrates, respectively. Thus, DprA*_Bsu_*-RecA*_Bsu_* interplay with a 20–21 long oligo will be further analyzed.

Co-crystal structures showed a binding footprint of 5-nt per DrpA*_Hpy_* monomer and 3-nt per RecA monomer [14,28], and physical interaction between DprA*_Bsu_* and RecA*_Bsu_* has been demonstrated (see above). Based on this, we propose that DprA*_Bsu_* bound to (dT)_20_ ssDNA interacts with and recruits RecA*_Bsu_*·dATP, contributing to the loading of ~5 RecA*_Bsu_*·dATP protomers. The DprA*_Bsu_*-RecA*_Bsu_* interaction may elicit an allosteric effect on both proteins. Under these conditions, monomeric DprA binds 5-nt ssDNA and enhances RecA*_Bsu_* cooperativity, facilitating dynamic filaments. This hypothesis aligns with previous findings that a DprA*_Spn_* monomer has only a 2.6-fold weaker binding affinity for (dT)_20_ compared to the DprA*_Spn_* dimer [13]. DprA*_Spn_* or DprA*_Bsu_* is sufficient to recruit homologous or heterologous RecA, in the ATP or dATP bound form, onto short ssDNA oligos [2,5,7].

### 2.5. Structural Model of the RecA-DprA-ssDNA Complex

Our dATP hydrolysis assays established that DprA*_Bsu_* caps and stabilizes RecA/RecA*_Bsu_* filaments on a (dT)_21/20_ substrate, implying that only one DprA*_Bsu_* DNA binding site engages each short ssDNA. When a DprA*_Bsu_*–RecA*_Bsu_*·dATP complex binds to a (dT)_20_ ssDNA, it moderately enhances dATP hydrolysis along the filament and yields RecA*_Bsu_*·dADP. This interaction appears sufficient to allosterically stabilize RecA*_Bsu_*, promoting cooperative ssDNA binding and the formation of helical nucleoprotein filaments. The N-terminal helix of RecA*_Bsu_* may destabilize the DprA*_Bsu_* dimer, favoring formation of a functional monomer, as suggested earlier [4,13]. This mechanism provides a molecular explanation for the transfer of short (dT) ssDNA from DprA*_Bsu_* to RecA*_Bsu_*. This hypothesis led us to refine an existing structural model of the complex to better align with the experimental data (Figure 4A,B).

Using molecular modeling approaches based on three different incomplete complexes of highly homologous proteins, RecA, DprA, and ssDNA, we built the fully atomic structure of the DprA_2_-RecA_2_·ATP-(dT)_15_ complex (see Section 3). Specifically, we combined (1) the structure of the presynaptic nucleoprotein filament of *Escherichia coli* RecA protein bound to ssDNA (PDB ID: 3CMX), (2) the DprA–ssDNA complex from *Helicobacter pylori* (PDB ID: 4LJR), and (3) a fully atomic spatial model of the *Streptococcus pneumoniae* RecA-DprA dimer generated by molecular docking [13,14,28]. All specific interactions observed in the original structures were preserved in the reconstructed DprA_2_-RecA_2_·ATP-(dT)_15_ complex (Figure 4). To connect the DNA segments from the RecA-DNA (PDB ID: 3CMX) and DprA-DNA (PDB ID: 4LJR) structures, we inserted two additional nucleotides. The specific interactions of this new structural element with the DprA and RecA proteins are shown in Figure 4B.

Although this nucleoprotein complex has a multicomponent structure, obtained from the structural data of complexes of various compositions, the structure and conformation of its constituent proteins turned out to be compatible with each other and do not have any structural problems. However, the conformation of the 5′-end of the ssDNA presynaptic filament RecA (shown in brown) turned out to be incompatible with the presence of DprA proteins and directed towards the region of DprA protein–protein interface and not towards the known nucleotide binding site found in the crystal structure of the dimeric complex of DprA proteins with short fragments of ssDNA [14]. This obviously means that binding of DprA requires that the six nucleotide residues of the free 5′-end of ssDNA adopt a suitable conformation for binding to the nucleotide binding site of the DprA protein. Molecular modeling of the conformational transition of ssDNA showed that the nucleotide from the dimeric complex of the DprA protein [14] and ssDNA of the presynaptic RecA protein filament [28] can be connected without any steric strains by two mobile nucleotide residues, thus forming a continuous ssDNA chain running from the nucleotide binding site of the DprA protein to the RecA proteins helical filament, as shown in Figure 4. This fully atomic model reconciles our biochemical data with structural constraints, providing a molecular basis for DprA’s one-site mediation of RecA loading on short ssDNA stretches.

The ATP/dATP hydrolysis site of RecA*_Bsu_* is located at the protomer–protomer interface. Filament assembly on ssDNA requires direct contacts between loop L1 of one RecA protomer to contact L2 of its neighbor protomer every three nucleotide residues. Hence, each additional RecA*_Bsu_* subunit in the nucleoprotein filament engages three nucleotides of ssDNA [28]. From this requirement, one ATP-binding site (two RecA*_Bsu_*protomers) demands at least 15 nt of ssDNA, whereas two sites (three protomers), which support higher ATP hydrolysis rates, require 18–20 nt. Our dATP hydrolysis assays with DprA*_Bsu_*-stimulated RecA*_Bsu_* filaments on (dT)_21_/(dT)_20_ substrates closely match these theoretical thresholds (Figure 2 and Figure 3).

Co-crystal structures of DNA complexes of DrpA*_Hpy_* and RecA report footprints of 5 nt per DprA*_Hpy_* monomer and 3 nt per RecA*_Hpy_* protomer [14,28], and we have shown that DprA*_Bsu_* physically interacts with RecA*_Bsu_*.

One important question concerns the role of the second subunit of the DprA protein in the DprA_2_–RecA_2_·ATP–(dT)_15_ complex shown in Figure 4. It is evident that an identical symmetrical structure could theoretically be formed through the interaction between the second DprA*_Bsu_* subunit and a RecA*_Bsu_* protein. However, the simultaneous presence of both symmetrical structures within a single complex is sterically hindered, as the two symmetrically located RecA*_Bsu_* subunits compete for the same spatial location.

Recent structural data and molecular modeling of DprA complexes with ssDNA and RecA have led to the proposal of a molecular mechanism for DprA-mediated loading of RecA*_Bsu_* onto ssDNA [29]. This model suggests that DprA forms a dimeric nucleoprotein complex with ssDNA, providing a platform for RecA nucleation. Following this initial interaction, one DprA subunit dissociates, allowing RecA filament formation on ssDNA to proceed. According to the model, ssDNA binds simultaneously to both DprA subunits, whose DNA binding sites are approximately 50 Å apart, equivalent to a stretch of ssDNA about 10 nucleotides long in a fully extended conformation.

Experimental analysis of DprA–ssDNA binding indicates that stability depends on the DNA length [14]. Short ssDNA fragments (6–20 nucleotides) either do not bind or have very low affinity, whereas high-affinity binding was observed for ssDNA fragments around 35 nucleotides long. The structure of the dimeric DprA-DNA complex was resolved using X-ray crystallography. However, most of the (dT)_35_ ssDNA appeared disordered, and electron density was observed only for the 6-nt at the 5′ end. These 6-nt were used in the reconstruction of the DprA_2_-RecA_2_·ATP-(dT)_15_ complex shown in Figure 4. RecA interacts with and destabilizes the DprA dimer interface, facilitating the transfer of ssDNA from DprA to RecA.

Importantly, due to the symmetry of the DprA dimer, its two ssDNA binding sites are oriented in opposite directions, making it sterically impossible for a single ssDNA strand to occupy both sites of the DprA dimer simultaneously. Consequently, only a complex in which ssDNA binds to one DprA subunit (Figure 4) is sterically feasible. The second DprA subunit, however, can accommodate a separate ssDNA strand in the opposite orientation. In this scenario, the second subunit not only promotes initial loading of the first RecA monomer onto ssDNA but may also play a central role in the still poorly understood mechanism by which the RecA nucleoprotein filament efficiently searches for homologous DNA. Furthermore, the binding of ssDNA to the second subunit could modulate the ssDNA binding affinity of the first. Under these conditions, the minimum ssDNA length required is estimated at 6 + 9 = 15-nt nucleotides, a value that aligns with the experimental data obtained in this study.

## 3. Materials and Methods

### 3.1. Strains and Plasmids

The donor strain KL227 (HfrP4x *metB*) and recipient strains AB1157 (*thr-1 leuB6 ara14 proA2 hisG4 argE3 thi-1 supE44 rpsL31*) and recombination-deficient JC10289 (genetically identical to AB1157 but carrying *ΔrecA-srlR306:Tn10* [*=ΔrecA306*]) were obtained from A.J. Clark’s collection. The strain JW5708-1 (*ΔdprA-724::kan*) was sourced from the Keio Collection (*E. coli* Genetic Resources at Yale CGSC). A null *dprA* (Δ*dprA*) strain was generated via P1 transduction, transferring the *dprA*-724 (del)::kan allele from JW5708-1 into AB1157, as described previously [30]. The pT7 plasmid (originally named pT7POL26) encodes T7 RNA polymerase under the control of a *lacZ* promoter. It was used to drive expression of DprA/Smf proteins via IPTG induction of the *lac* promoter. Plasmid pCB888, which contains the *dprA_Bsu_* gene under the control of a T7 RNA polymerase-driven promoter, was kindly provided by Prof. J.C. Alonso. Likewise, the pDprA plasmid, harboring the *dprA* gene with a similar promoter structure, was supplied by EUROGEN (Moscow, Russia).

### 3.2. Proteins

Wild-type RecA*_Eco_* and RecA*_Bsu_* were purified as previously described [31,32]. Single-strand binding (SSB) protein was kindly provided by Prof. M. Cox (University of Wisconsin–Madison). Protein concentrations were determined using their native extinction coefficients: ϵ_280_ = 2.23 × 10^4^ M^−1^ cm^−1^ for RecA [33], 1.52 10^4^ M^−1^ cm^−1^ for RecA*_Bsu_*, and 2.38 × 10^4^ M^−1^ cm^−1^ for SSB protein [34]. The concentration of DprA*_Bsu_* was determined using the native extinction coefficient ϵ_280_ = 4.5 × 10^4^ M^−1^ cm^−1^ [5].

DprA*_Bsu_* purification was performed following previously reported protocols [5], with some modifications. Briefly, *E. coli* BL21(DE3) pLysE cells were transformed with the pCB888. Cultures were grown at 25 °C, and protein expression was induced with 0.4 mM isopropyl-1-thio-β-D-galactopyranoside (IPTG) at an OD_600_ of 0.5. After induction, cells were incubated for 3 h, with rifampicin (200 µg/mL) added 30 min after IPTG induction.

DprA*_Eco_* was purified from the clarified lysate via two-step chromatography using HisTrap HP 1 mL (GE Healthcare, Chicago, IL, USA) and HiTrap SP XL 1 mL columns (GE Healthcare, Chicago, USA). Protein-containing fractions were concentrated using Amicon Ultra-4 centrifugal filter (Merck, Darmstadt, Germany) with a 3 kDa cut-off, supplemented with 50% glycerol, and stored at −20 °C.

### 3.3. (d)ATP Hydrolysis Assays

A coupled enzyme spectrophotometric assay was employed to measure RecA-mediated ATP or dATP hydrolysis, as previously described [35,36]. In this system, the ADP or dADP generated by hydrolysis was converted back to ATP or dATP by a regeneration system involving pyruvate kinase and phosphoenolpyruvate (PEP). The resulting pyruvate was subsequently converted to lactate-by-lactate dehydrogenase, using NADH as a reducing agent. This reaction enabled indirect monitoring of nucleotide hydrolysis via the decrease in absorbance at 380 nm due to the oxidation of NADH to NAD^+^. ATP or dATP consumption over time was quantified using the NADH extinction coefficient ε_380_ = 1.21 mM^−1^ cm^−1^. All measurements were conducted using a Cary 5000 dual-beam spectrometer equipped with a temperature controller and a 12-position cell changer (Varian, Palo Alto, CA, USA). The path length was 1 cm, and the band pass was 2 nm. RecA*_Eco_* ATPase and dATPase assays were performed in buffer A, composed of 25 mM Tris-HCl (pH 7.5, 88% cation), 10 mM MgCl_2_, 5% *w/v* glycerol, 1 mM dithiothreitol (DTT), and 2 mM ATP or dATP. The regeneration system included 3 mM PEP, 10 U/mL pyruvate kinase, 10 U/mL lactate dehydrogenase, 4.5 mM NADH, and 5 μM M13mp18 cssDNA or (dT) oligos. RecA*_Bsu_* dATPase assays were performed using buffer B, which consisted of 50 mM Tris-HCl [pH 7.5], 10 mM Mg(OAc)_2_, 5% *w/v* glycerol, 1 mM dithiothreitol (DTT), and 2.5 mM dATP. The regeneration components included 0.5 mM PEP, 10 U/mL pyruvate kinase, 10 U/mL lactate dehydrogenase, 4.5 mM NADH, and 10 μM (dT)_n_ oligos. All reactions were repeated at least in triplicate, yielding consistent results.

### 3.4. Conjugation

Conjugation experiments were carried out essentially as previously described [16]. Expression was induced for no longer than 1 h prior to conjugation to limit cellular stress and maintain near-physiological protein levels. Both Hfr and F^−^ strains were cultured, mated, and recombinant progeny were selected at either 37 °C or 30 °C (as specified in the Table 1) for 1 h in mineral salts 56/2 medium supplied with all necessary growth factors at pH 7.5. The donor-to-recipient ratio in the mating mixture was 1:10, with 2–4 × 10^7^ donors and 2–4 × 10^8^ recipients per 1 mL. The yield of *thr*^+^ Str^r^ and *ara*^+^ Str^r^ recombinants from independent crosses (typically 5–7% relative to donor input) was normalized to account for the mating ability of each recipient strain. This was determined by measuring the yield of F’-lac^+^ transconjugants in control crosses between recipient and donor strains P200 F’-lac.

FRE (frequency of recombination exchange) values were calculated according to previously established procedures [16]. Changes in FRE (ΔFRE) resulting from *recA* mutation or accessory gene, relative to the FRE promoted by the wild-type *E.coli recA,* were computed using the following formula: ΔFRE = ln(2μ_1_ − 1)/(2 μ_2_ − 1), where μ_1_ denotes the linkage of selected thr^+^ or ara^+^ with unselected leu^+^ markers in crosses involving the wild-type *E. coli* strain AB1157, and μ_2_ corresponds to the linkage observed in the experimental cross. Uncertainty in relative FRE values was calculated as the deviations from the average value based on three independent experiments, using the formula [=2 × STDEV] in Excel-97 and entering replicate data points.

### 3.5. Reconstruction of the Structure of the RecA-ssDNA-DprA_Bsu_ Complex

To construct a full-atom model of the RecA-ssDNA-DprA*_Bsu_* spatial structure, we utilized several crystal structures from the Protein Data Bank. These included the presynaptic nucleoprotein filament of RecA*_Eco_* bound to RecA_6_–(ADP–AlF_4_–Mg)_6_–dT_18_ (PDB ID: 3CMX) [28], the *Helicobacter pylori* DprA*_hpy_*–ssDNA dimer complex (DprA*_Hpy_*–dT_6_)_2_ (PDB ID: 4LJR) [14], and a molecular docking-derived spatial model of the RecA–DprA complex, kindly provided by the authors of this study [13]. Superimposition, energy minimization, and other molecular manipulations were performed using standard protocols of the ICM-Pro software package version 3.8-7c (Molsoft LLC, San Diego, CA, USA) [37], employing the ECEPP/3 force field as implemented in ICM-Pro [37,38].

The spatial structure of the DprA dimeric complex from *E. coli* was derived via homology modeling based on the DprA*_hpy_* crystal structure (PDB ID: 4LJR), using built-in modeling protocols in ICM-Pro. The RecA–sDNA–DprA*_Bsu_* complex was assembled through a stepwise superimposition of individual protein structures. Initially, the position of the DprA*_Bsu_* dimer was determined by aligning RecA monomers from the presynaptic filament (PDB ID: 3CMX) [28] and in the docking model of the RecA–DprA*_Bsu_* complex [13]. Next, the location of 6-nt at the 5′-end of ssDNA was defined by superposing the (DprA*_Hpy_*-dT_6_)_2_ complex (PDB ID: 4LJR) [14] with the RecA–DprA model. The conformation of the 3′ ssDNA segment was adopted from its positioning within the RecA presynaptic filament (PDB ID: 3CMX) [28]. Finally, the two bridging nucleotides were modeled by minimizing the energy of the ssDNA strand, keeping its 5′ and 3′ends fixed.

## 4. Conclusions

In this study, we investigated the functionality of the DprA*_Eco_* homolog in the context of Hfr conjugation. While most regulators only partially influence the genetic outcomes of recombination, thereby merely modulating the natural FRE values (used to assess in vivo recombinase activity) [16,18,39,40], our study identifies DprA/Smf as a potent activator of recombination, increasing FRE by approximately 18-fold, upon overexpression. This finding suggests that the DprA–RecA interaction in *E. coli* represents an additional layer of control over the primary recombination pathways and advances our understanding of homologous recombination regulation. Collectively, our data indicate that DprA proteins may play distinct roles across different bacterial species.

Through length-dependent experiments, we determined that a 20-nucleotide ssDNA substrate is sufficient to nucleate the RecA*_Eco_* filament in conjunction with DprA*_Bsu_*. This led us to refine an existing structural model to better align with our biochemical data.

Building on previous mutagenesis-based analysis [4], which showed that DprA*_Spn_* dimerization is essential for DNA binding and RecA*_Spn_* loading, and consistent with data from *H. pylori* demonstrating that only the dimeric form is active [14], we propose a new mechanism.

Structural and biochemical characterization of the DprA proteins reveals that a DprA_2_–ssDNA complex forms with only one DprA subunit actively bound to ssDNA, while the second DprA subunit sterically stabilizes the initial RecA monomer in its DNA-bound conformation. Based on the known dimensions of the DprA–DNA binding site and the role of dimeric DrpA as the active species, we propose the formation of a ternary RecA–DprA–ssDNA complex during RecA filament nucleation. Within this configuration, the remaining 21 nt ssDNA stretch can accommodate, at most, a RecA trimer as the minimal nucleation unit, which then extends into a hexamer upon DprA dissociation.

This structural model of the DprA-ssDNA complex provides a compelling basis for further investigation into DprA and related proteins. Regulators of homologous recombination involved in chromosomal transformation should undoubtedly be considered promising targets for inhibition [41,42]. Further structural analysis of the DprA–RecA complex may ultimately support the rational design of compounds aimed at suppressing horizontal gene transfer.

## Figures and Tables

**Figure 1 ijms-26-07873-f001:**
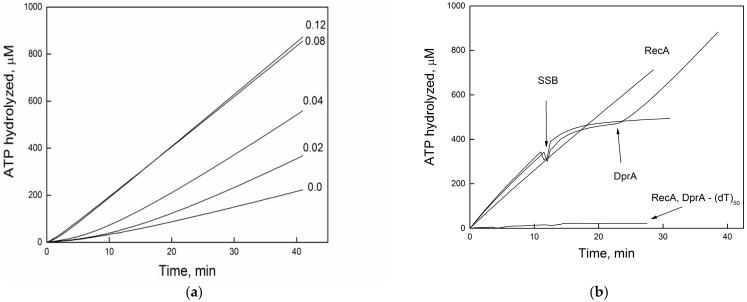
DprA*_Bsu_*antagonises SSB*_Eco_* function. (**a**) DprA*_Bsu_* reverses SSB*_Eco_* inhibition of *RecA_Eco_* ATP hydrolysis. Reactions mixtures were carried out as described under “Experimental Procedures” and contained circular M13mp8 ssDNA (12 μM in nt), RecA*_Eco_*(1 μM), SSB*_Eco_* (0.5 μM), and the indicated concentrations of DprA*_Bsu_*. (**b**) RecA (3 μM) was preincubated with poly(dT)_50_ (5 μM in nt). When indicated, SSB*_Eco_* (0.5 μM) or DprA*_Bsu_* (80 nM) was added. All assays were performed at least three times with consistent results.

**Figure 2 ijms-26-07873-f002:**
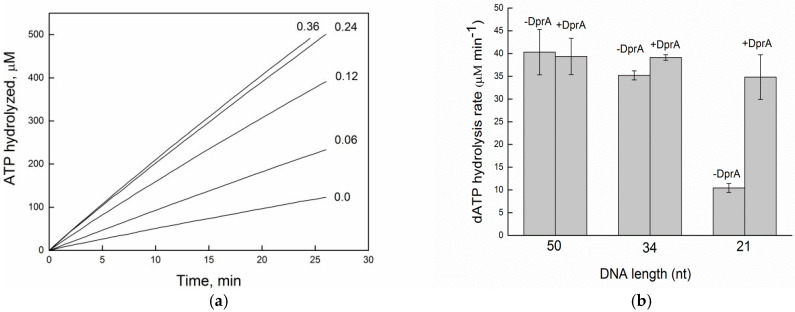
DprA_Bsu_ enables RecA*_Eco_* nucleation on short (dT)_n_ oligonucleotides. (**a**) ATPase activity of RecA*_Eco_* RecA (3 μM) on (dT)_34_ (5 μM in nt) in the presence of 2 mM ATP and increasing concentration of DprA*_Bsu_*. (**b**) dATPase activity of RecA*_Eco_* (5 µM) on (dT)_n substrates (5 µM in nt), with or without DprA*_Bsu_* (0.3 µM) and 2 mM dATP. All experiments were carried out at least three times with consistent results. All reactions were performed as described under “Experimental Procedures. Data are mean ± SD from three independent experiments.

**Figure 3 ijms-26-07873-f003:**
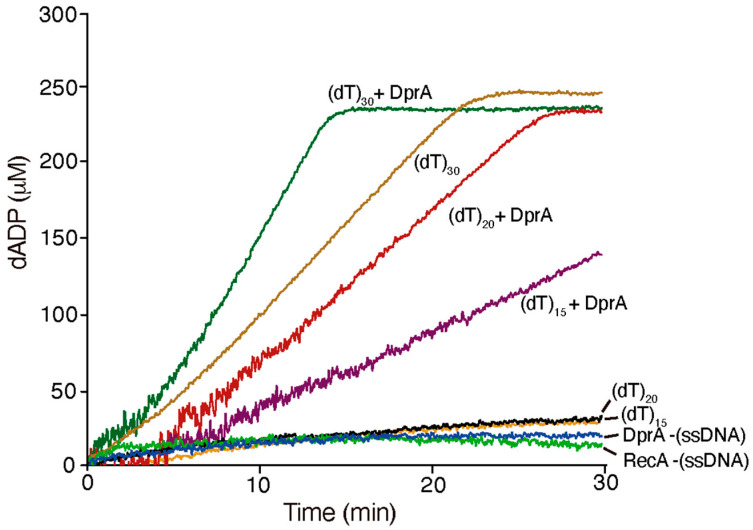
Effect of DNA size on RecA_Bsu_-dependent dATP hydrolysis in the presence or absence of DprA_Bsu_ protein. Reaction mixtures contained 3 μM RecA_Bsu_, 10 μM (dT)_n_ (in nt), and 2.5 mM dATP in Buffer B. When indicated, 0.3 μM DprA_Bsu_ was added. The dATP hydrolysis rate (K_cat_) was determined as described in Section 3.3.

**Figure 4 ijms-26-07873-f004:**
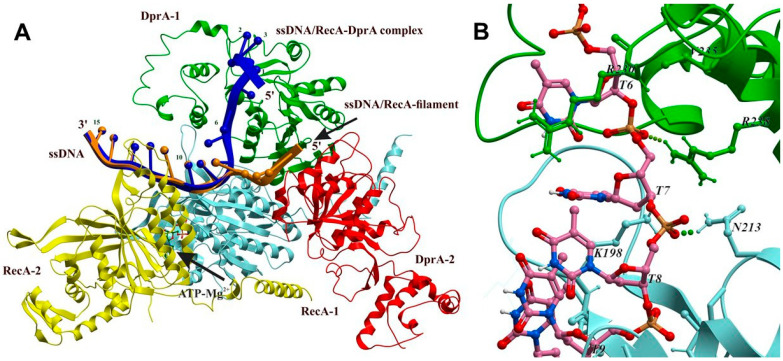
Atomic model of the DprA_2_-RecA_3_·ATP-(dT)_15_ complex. Panel (**A**) shows DprA*_Bsu_*, with protomers shown in green and red, and the two RecA*_Bsu_* protomers are shown in cyan and yellow, respectively. The ssDNA (dT_15_) strand bridges both proteins; the first six 5′ terminal nucleotides (blue) adopt the DprA-bound conformation (PDB ID: 4LJR) [13], while the seven 3′ terminal nucleotides follow the RecA filament path (PDB ID: 3CMX) [28]. The conformations of the two bridging nucleotides (T7 and T8) were obtained using molecular modeling in this study. The original RecA nucleoprotein filament (brown) is overlaid for reference. ATP–Mg^2+^ at the RecA interface is shown in the ball-and-stick representation. Panel (**B**) provides a close-up of the DNA-bridging region, highlighting interactions with DprA*_Bsu_* (green) and RecA*_Bsu_* (cyan). PDB formatted file is presented in the Appendix A.

**Table 1 ijms-26-07873-t001:** (**a**) The dependence of FRE value on the elevated DprA expression in transconjugants of the AB1157 strain crossed with the KL227 donor at 37 °C. (**b**) The dependence of the FRE value on elevated expression of DprA*_Bsu_* in transconjugants of the AB1157 strain crossed with the KL227 donor at 30 °C.

Expression Plasmids	Linkage (μ) Between Selected *thr*^+^ and Unselected *leu*^+^ Markers	Yield of *thr*^+^ Str ^R^ Recombinants (% to Donors)	FRE	ΔFRE
(**a**)				
ABpT7, Δ*dprA*pET21	0.93 ± 0.025 (600) *	1.6%	4.5 ± 1.0	1.0
ABpT7, Δ*dprA*pDprA	0.84 ± 0.032 (600)	0.71%	11.7 ± 1.5	2.6
ABpT7, Δ*dprA*pDprA*_Bsu_*	0.69 ± 0.021 (400)	0.22%	28.9 ± 5.3	6.4
(**b**)				
ABpT7, Δ*dprA*pET21	0.949 ± 0.024 (600) *	1.7%	3.2 ± 0.6	1.0
ABΔ*dprA*, pET21	0.94 ± 0.018 (600)	3.1%	3.6 ± 0.5	1.1
ABpT7, Δ*dprA*pDprA	0.59 ± 0.025 (800)	0.2%	55.5 ± 6.3	17.3
ABpT7, Δ*dprA*pDprA*_Bsu_*	0.56 ± 0.036 (600)	0.02%	61.2 ± 8.5	19.1

* Numbers in parentheses indicate the number of clones analyzed.

## Data Availability

The data that support the findings of this study are available on request from the corresponding authors.

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
