# Peer review of "A Single DNA Binding Site of DprA Dimer Is Required to Facilitate RecA Filament Nucleation"

_ijms, 2025, doi:10.3390/ijms26167873_

Round 1

Reviewer 1 Report

Comments and Suggestions for Authors

The manuscript by Bakhlanova et al “A single DNA binding site of DprA dimer is required to facilitate RecA filament nucleation“ describes transformation activity of DprA proteins from B. sub. and E. coli and in vitro activity of DprA to load RecA on SSB-bound and short oligonucleotides. The authors proposed the 3D model of the RecA-DprA-ssDNA complex based on published structures and their data.

Overall, some results can be potentially interesting and novel, thus advancing the understanding of DprA mechanism. However, the paper is poorly written with numerous grammar and syntax errors, missing words, incomplete sentences and multiple repetitions. References to panels a) and b) in Figs 1 and 2 are either mixed up in the text and legends or simply absent (what are numbers in Fig 2a?).  Introduction does not clearly outline outstanding questions in the field mixing up transformation activities, differences between remote homologs and biochemical activities. The presentation should be significantly improved before consideration for proper review and publication.

Specific comments:

  1. What is the significance of section 2.1? Neither introduction nor result section clearly describes missing knowledge about well-known transformation role of DprA. How conserved Ec and Bs sequences? What is the difference between “in vivo recombinase activity” and “normal conditions” (Discussion).
  2. The novelty and significance of results 2.2 are not clear. Was the recombination mediator activity of DprA in the presence of SSB not assayed before? Is the goal to assays heterospecific interactions? In such a case, it is important to show how activities of Bs and Ec DprA are related? The title of this section should not be general.
  3. Results in Fig 1 alone do not provide sufficient information for speculations about different mechanisms outlined at the end of section 2.2.
  4. Please, keep in mind during discussion that RecA nucleation unit is not the same as an ATPase efficient filament. Nucleation can be efficiently initiated by dimer or pentamer, however, neither of these can yield measurable ATPase activity which is linked to dissociation. Likewise, ATPase activity may not efficiently report binding of RecA monomer and short filaments to DprA/ssDNA.
  5. Are molarities of oligo in Fig 3 in nucleotides or molecules? Should the difference in ATPase rate be adjusted in the latter case or at least mentioned that lower rate may be related to fewer RecA bound to shorter DNA?
  6. The proper control for Fig 3 should be RecA with DprA in the absence of DNA (unless already published).
  7. Description of existing structures used in modeling should be clearer and more detailed, otherwise, it is very difficult to follow the logic of modeling.
  8. The authors should use dT15 for modeling and discussion, since ATPase activity is detected for dT15 in the presence of DprA.
  9. The modeling should be accomplished by explicit description of known DNA and RecA binding sites of DprA and whether specific amino acids involved in interaction are conserved. Mapping key binding elements suggested by the model on the sequence alignment will be very helpful. Since experiments were performed with heterospecific interactions, such alignment figure may provide a better proof for the model.
  10. There is no discussion of the published model suggested dissociation of DprA dimer upon interaction with RecA.

Reviewer 2 Report

Comments and Suggestions for Authors

The ms by Bakhlanova et al describes mostly biochemical experiments with the natural transformation protein Dpr1, which lead to the conclusion that a single binding site of a DprA dimer is required for RecA nucleation. Also they show that extra DprA increase the frequency of Hfr mediated exchanges. I have the following remarks:

The introduction is somewhat repetitive and could be more concise. Also, from line 79 the text is no longer introduction, but a mix of methodology and summary, and could be omitted in my opinion.

Line 120- please, better explain ΔFRE. FRE1 is .., FRE2 is..

Line 124- describe how time of protein overexpression was adjusted. Why weren't these experiments done in a dprA background to avoid potential difficult-to-foresee interactions between DprA(Eco) and DprA(Bsu)

Line 138- Apparently FRE values are higher at 42C than at 37C, While in the current experiments FRE values at 30C were much higher than at 37C. Itn't it likely then that another factor could be responsible for the 42C effect seen in the previous experiments?

Discussion Recently, it was published by Dalia et al (PNAS 122, e2421764122) that DprA from Gram negative bacteria has a C-terminal domain with which it interacts with ComM to promote branch migration. This could be discussed in the ms. Does E. coli lack a ComM ortholog explaining lack of clear difference between the Ec and Bs DprA's in the current experiments?

Reviewer 3 Report

Comments and Suggestions for Authors

This is a solid and timely manuscript, with interesting data on the role of B. subtilis DprA in facilitating RecA nucleation on short ssDNA substrates. The experimental design is generally sound, and the integration of ATPase assays with a molecular model is a strength. However, there are a few concerns that should be addressed before the manuscript is considered for publication.

Please, state the limitations of your data shown in Figure 4. The model is purely computational, and there's no supporting structural evidence—no co-crystallization, mutagenesis, or biochemical mapping to confirm the interface. 

The ATPase/dATPase rates shown in the figures are barely above background in many conditions, especially with shorter oligos. The interpretation that RecA is nucleating onto ssDNA in these cases is not fully convincing unless appropriate negative controls are shown. Specifically, you should include RecA and DprA alone (without DNA) for each (dT)n tested, to rule out basal activity. 

There is also no discussion about a potential function of monomeric DprA. Since your model relies on only one DNA-binding site of the dimer, it would be relevant to at least comment on whether monomeric forms could contribute in any way, even transiently. 

Finally, the use of a RecA mutant defective in filament formation (e.g. impaired in L1/L2 loops or N-terminal interactions) would greatly strengthen your conclusion that the increased hydrolysis is due to filament assembly. Even if not feasible now, this should be mentioned as a limitation.

Minor comments:

Species names should be italicized consistently throughout the abstract and main text. Streptococcus pneumonia should be corrected to Streptococcus pneumoniae.

Lines 43–45: this sentence is too long for a relatively minor clarification. You could simply introduce an 'Ec' prefix for all E. coli genes and proteins to improve clarity.

Line 56–57 is difficult to follow. Please revise for clarity.

Avoid informal abbreviations like "aka"—this is not standard in academic writing.

All Latin-derived expressions (e.g. in vivo, in vitro) should be in italics.

Lines 74–76 and 79 need rephrasing.

The final two paragraphs of the introduction should be merged and introduced with “In this study” to make it explicit that you are referring to your own data and not general background.

Tables 1a and 1b use different conditions without a clear rationale. If the point is to compare different induction temperatures, this should be explained more explicitly, and the tables could be merged into a single one.

Lines 138–139: you state that the result is surprising, but then say it was previously described, which is contradictory.

Lines 152–153 also need rephrasing.

Line 158: "as observed in the referenced study" is too vague. It would be clearer to say "as previously reported [ref]" and to clarify whether these are new or replicated observations.

Line 173: avoid vague references to "Experimental Procedures" unless a specific subsection is named.

Figure 1 lacks indications of variability. Please add error bars or shaded confidence intervals to show the degree of reproducibility.

Line 188: references are needed here to support the statements on RecA/SSB dynamics.

Overall, the Results section reads more like a mixed results and discussion section. 

Line 223: the use of "anyway" is informal and should be avoided.

Line 293: same issue as with line 173—be specific or remove the pointer to "Experimental Procedures."

Line 407 also needs rephrasing.

The current "Discussion" section is mostly a summary of the findings and functions more like a Conclusion. 

Finally, please check species name formatting in the bibliography as well, they are not italicized.

Round 2

Reviewer 1 Report

Comments and Suggestions for Authors

The authors considerably improved manuscript. There are still numerous errors in text, some of which are noted below; however, results and conclusions are well-described and clearly presented.

  1. The major comment of this reviewer is to the modeling and its interpretation (thank you for providing coordinates): In the model, DprA interacts only with 3 nucleotides. Three 5’ Ts are in solution and do not interact with proteins. In addition, two 3’ Ts are not bound to RecA either. Therefore, one DprA monomer and two RecAs engage only 10 nucleotides including the linker region. Thus, 15 nt ssDNA can potentially accommodate one DprA and 3-4 RecAs in the current conformation. The authors should reconsider and alter the model by removing unbound nucleotides from 5’ end and extending 3’ end with adding additional RecA monomers (this will not require additional minimization, since adding more RecA monomers from known crystal structure will not affect RecA-DprA interface). This will considerably strengthen their conclusions and experimental results, including the observed dATP hydrolysis on dT15.
  2. The model further support and explain the measurable dATP hydrolysis rate with dT15 in Fig 3, and 15-mer should be considered as a minimal unit.
  3. The reason for switching from Eco to less active Bsu RecA is not clear, however, results still support major conclusion.
  4. The notion of 2 nt binding site for Bsu RecA does not make sense.
  5. The control of RecA w/o SSB and DprA , if available, will be beneficial in Fig 1a).
  6. It would be great to test DprA mutant which do not form dimer for RecA stimulation, however, this is likely beyond the scope of the current manuscript.

Minor comments and notes of some errors in the text:

Line 18: Is it EcRecA?

Lines 18-22:

The sentence should be altered to something like “In vitro, B. s. DprA reverses the inhibitory effect of SSB on RecA …

Line 36: “ssDNA” is defined in previous sentence.

Line 45: …, cellular function of the protein [].

Line 48: replace catalysis with “promoting” or “accelerating” or similar.

Line 49: counteract function of negative RecA modulator RecX

Lines 72-73: fix typos.

Line 78: “tool protein”?

122: “respectively”?

123: How “unexpected trend” can “underscores the robustness”?

130: should be “poor solubility” or “insolubility”

137: “), go” -> “to”

138: Why 1 uM RecA is sub-stoichiometric to 0.5 uM SSB? Only DprA is in substoichiometric amount in this reaction.

250: conclusion about minimal site for mediator assistant nucleation on 21-nt ssDNA is not valid since shorter lengths were not tried (or shown) in Fig 2b.

273: DprA_Bsu

275-276, 277-278: Fix sentences.

280: between?

295: cooperative binding?

351: “Co-crystal structures” suggest a crystal structure of DprA-RecA-ssDNA complex. It should be rephrased as structures of DNA complexes of DprA and RecA.

 415: c

The entire text should be subjected to spelling and punctuation check to avoid typos and errors.

Reviewer 3 Report

Comments and Suggestions for Authors

No further comments.

Author Response

No further comment. The Reviewer raised no new issues in this round. We thank the reviewer and the editor for their consideration.